# Responses of *Acidithiobacillus thiooxidans* A01 to Individual and Joint Nickel (Ni$^{2+}$) and Ferric (Fe$^{3+}$)

**Aijia Chen [1,2], Xiaodong Hao [1,2], Yunhua Xiao [1,2], Kai Zou [1,2], Hongwei Liu [1,2], Xueduan Liu [1,2], Huaqun Yin [1,2], Guanzhou Qiu [1,2] and Yili Liang [1,2,*]**

[1] School of Minerals Processing and Bioengineering, Central South University, Changsha 410083, China; 17136415094@163.com (A.C.); haoxiaodongxyz@163.com (X.H.); huazipiaoling.123@163.com (Y.X.); zoukai3412085@csu.edu.cn (K.Z.); hongweiliu@csu.edu.cn (H.L.); xueduanliu@yahoo.com (X.L.); yinhuaqun_cs@sina.com (H.Y.); qgzhoucsu@163.com (G.Q.)

[2] Key Laboratory of Biometallurgy of Ministry of Education, Changsha 410083, China

[*] Correspondence: liangyili6@csu.edu.cn; Tel.: +86-731-8883-0546

**Abstract:** *Acidithiobacillus thiooxidans* A01 is widely used in bioleaching processes and commonly thrives in most metal-rich environments. However, interactions between different heavy metals remain obscure. In this study, we elaborated the effect of ferric iron on the growth and gene expression of *At. thiooxidans* A01 under the stress of nickel. The results showed that 600 mM Ni$^{2+}$ completely inhibited the growth and sulfur metabolism of *At. thiooxidans* A01. However, trace amounts of Fe$^{3+}$ (0.5 mM) facilitated the growth of *At. thiooxidans* A01 in the presence of 600 mM Ni$^{2+}$. With the addition of 5 mM Fe$^{3+}$, the maximum cell density reached $1.84 \times 10^8$ cell/mL, and pH value was 0.95. In addition, metal resistance-related and sulfur metabolism genes were significantly up regulated with extra ferric iron. Taking the whole process into account, the promoting effect of Fe$^{3+}$ addition can be attributed to the following: (1) alleviation of the effects of Ni$^{2+}$ toxicity and restoring the growth of *At. thiooxidans* A01, (2) a choice of multiple pathways to export nickel ion and producing precursor of chelators of heavy metals. This can suggest that microorganisms may widely exhibit metabolic activity in iron-rich environments with heavy metals. Our study will facilitate the technique development for the processing of ore bodies with highly challenging ore compositions.

**Keywords:** *Acidithiobacillus thiooxidans* A01; nickel resistance; ferric iron; bioleaching

---

## 1. Introduction

Biohydrometallurgy is widely used in industrial bioleaching of ores to extract metals such as copper, nickel, gold, uranium. It is done mainly by acidophilic microorganisms [1]. Some of the heavy metals are essential for the metabolic activity of cells at low levels, while they exert toxic effects at concentrations encountered in polluted environments.

Nickel has been identified as a component in a number of enzymes, participating in important metabolic reactions such as ureolysis, hydrogen metabolism, methane biogenesis and acitogenesis [2]. In this way, microorganisms have evolved in the presence of this metal, which is necessary in trace amounts for a variety of metabolic processes but toxic in high concentrations, causing oxidative stress in the cell [3]. The toxicity of nickel is attributed to its replacement of metals in metalloproteins, to its binding to catalytic residues in sulfur dioxygenase, sulfite oxidase and the plasma membrane, and indirectly to its exertion of oxidative stress [4].

Several nickel resistance bacteria have been isolated mainly from heavy metal-contaminated samples [5]. To tackle Ni toxicity, microorganisms have developed diverse mechanisms including extracellular detoxification, intracellular sequestration, modification of cation transport systems and

active transport by efflux pumps [6]. Most nickel resistance determinants known to date are efflux pumps isolated from cultivable bacteria. The resistance-nodulation-cell division (RND) protein family was described as a related group of bacterial transport proteins involved in heavy metal resistance [7]. In Gram-negative bacteria, the Cobalt-zinc-cadmium resistance transporter (CzcCBA) (consisting of subunits C, B and A) is a member of the RND system that exports metals including cobalt, zinc and cadmium from the cytoplasm or the periplasm across the outer membrane [8]. The complex achieves heavy metal resistance by active cation efflux driven by a cation-proton antiporter [9,10]. In *Cupriavidus metallidurans* strain CH34, RND protein is central to the process of metal cation homeostasis that is adapted to high concentrations of $Zn^{2+}$, $Cd^{2+}$, $Ni^{2+}$, $Co^{2+}$, $Cu^+$ or $Ag^+$ [11].

New nickel resistance systems are still being discovered. A nickel hyperaccumulation mechanism is proposed for a serine O-acetyltransferase [12]. Seven different genes conferring Ni resistance were identified in *Acidiphilium* sp. PM. Two of them form an operon encoding the ATP-dependent HslVU (ClpQY) protease. Other Ni-resistance determinants include genes involved in lipopolysaccharide biosynthesis and the synthesis of branched amino acids and membrane fatty acid [13,14]. *Acidithiobacillus ferrooxidans* and *Pseudomonas* sp. MBR had the ability to reduce $Ni^{2+}$ to its elementary forms coupled to biomineralization under aerobic conditions [15,16]. It suggests that Ni resistance relies on different molecular mechanisms.

Nickel and other heavy metals are often encountered in an aquatic environment. Gikas observed that $Ni^{2+}$ and $Co^{2+}$ acted synergistically at the increasing stimulation and at the intoxication zones, whilst an antagonistic relation was determined at the decreasing stimulation zone [2]. Similar experiments implied that ferrous iron gave significant protection to *Acidithiobacillu caldus* BC13 against lead and zinc toxicity [17]. *Bacillus* sp. may widely exhibit catabolic activity in iron-rich environments with heavy metal [2]. It is obvious that interactions between different heavy metals remain obscure, while it should be taken into account in the methodologies used to establish criteria for tolerance levels in the environment.

*At. thiooxidans* is a conspicuous dweller of acidic metal-rich environments. It is well known for its rapid oxidation of elemental sulfur. It is able to oxidize sulfur and reduced sulfur compounds for energy but not $Fe^{2+}$. *At. thiooxidans* A01 is not able to tolerate high levels of $Ni^{2+}$. The co-culture of *At. thiooxidans* and iron-oxidizing *Leptospirillum ferriphilum* showed a stronger sulfur oxidation and ferrous oxidation activity under the stress of heavy metals including $Ni^{2+}$ and $As^{3+}$ [18,19]. It indicated that there were synergetic interactions between iron-oxidizing and sulfur-oxidizing bacteria. We suggested that the metabolites of *L. ferriphilum* may facilitate the growth of *At. thiooxidans* under the stress of nickel. In this study, we investigated the effect of ferric iron on the growth and sulfur metabolism of *At. thiooxidans* A01 in the presence of $Ni^{2+}$ to clarify the mechanism at transcription level.

## 2. Materials and Methods

### 2.1. Bacterial Strain, Medium and Growth Condition

In this study, the bacterial strain was *At. thiooxidans* A01 (FJ15-4526), which was isolated and identified by our laboratory from Ping Xiang coal mine [20]. It was grown in 9K medium, which contained 3.0 g/L $(NH_4)_2SO_4$, 0.5 g/L $MgSO_4 \cdot 7H_2O$, 0.5 g/L $K_2HPO_4$, 0.1 g/L KCl, 0.01 g/L $Ca(NO_3)_2$. The initial pH value of the medium was 1.8 and the medium was autoclaved for 20 min at 121 °C. The energy resource was elemental sulfur ($S^0$, 10 g/L) and it was incubated at 30 °C and 170 rpm.

### 2.2. Nickel Resistance Experiment with Different Ferric Iron Concentration

To investigate the effect of ferric iron on nickel resistance, this experiment was designed to form a ferric iron concentration gradient during *At. thiooxidans* A01 incubation with a certain nickel concentration. After exposure to ultraviolet light (10 W, 254 nm) for 30 min, ferric sulfate ($Fe_2(SO_4)_3 \cdot 7H_2O$; 0, 0.5, 2.5, 5 and 50 mM), nickel sulfate ($NiSO_4 \cdot 6H_2O$; 600 mM) and elemental

sulfur ($S^0$; 10 g/L) were added in previously autoclaved 100 mL 9K medium, which was in 250-ML Erlenmeyer flasks. The initial cell density was $1 \times 10^7$ cells/mL, the initial pH value of the medium was 1.8, and they were incubated in a shaker incubator set at 30 °C and 170 rpm. Each treatment was performed in triplicate.

The physicochemical parameters including pH and cell density were measured every day. The pH value was measured by pH meter, the density of the microorganism was counted by a Thomas chamber with an optical microscope at $100\times$ magnification.

### 2.3. Shock Treatment

The shock experiment was conducted to explore the influence of ferric iron or nickel on the relative gene expression. *At. thiooxidans* A01 was incubated with 10 g/L $S^0$ at 30°C and 170 rpm, and the initial cell density was $1 \times 10^7$ cells/mL and the initial pH of 9K medium was also 1.8. We designed three treatments, including 5 mM ferric iron shock, 600 mM nickel sulfate shock, and 5 mM ferric iron, 600 mM nickel sulfate joint shock. The shock treatment was at the 108th hour, the cell density was $1.00 \times 10^8$ cells/mL, and the pH was 0.77. The control group was not shock treated. Before utilizing, 9 K medium, $S^0$, ferric sulfate and nickel sulfate were sterilized. After shock treatment, cells were harvested for RNA extraction in the 10th, 30th, 60th and 120th mins, the cell densities were $1.00 \times 10^8$, $1.03 \times 10^8$, $1.06 \times 10^8$, $1.18 \times 10^8$ cells/mL in the presence of $Fe^{3+}$, $1.00 \times 10^8$, $7.95 \times 10^7$, $8.80 \times 10^7$, $4.51 \times 10^7$ cells/mL in the presence of $Ni^{2+}$, $1.00 \times 10^8$, $7.96 \times 10^7$, $8.80 \times 10^7$, $6.00 \times 10^7$ cells/mL in the presence of $Ni^{2+}$ and $Fe^{3+}$, respectively.

### 2.4. Cell Collection and RNA Extraction

The process of cell collection was shown in a previous study by Jiang et al. [20]. The cells were filtered out using a filter paper (0.45 μm) to remove the remaining sulfur. The solution was put into centrifuge bottle and centrifuged at 12,000 rpm for 20 min to collect the cells. The total RNA extraction was isolated and purified according to the method described by Wang et al. [8]. About 1 μg purified RNA from each sample was used for cRNA synthesis with the ReverTra Ace Qpcr RT Kit (Toyobo, Osaka, Japan), according to the manufacturer's protocol. RNA extracts were treated with DNase to remove DNA before cDNA synthesis.

### 2.5. Primers and Real-Time Polymerase Chain Reaction (PCR)

The genome of *At. thiooxidans* A01 has been sequenced, completely annotated and deposited at the DDGJ/EMBL/Genbank with the accession number AZMO0000 [21]. Nickel resistance-related genes and growth-related genes were predicted based on the genome sequences. Specific primers used in this study are shown in Table 1. The specificity of primers was checked by conventional polymerase chain reaction (PCR) and sequencing.

**Table 1.** Specific primers used for polymerase chain reaction (PCR) and real-time PCR.

| Primer Name | Target Gene | Sequences (5′–3′) | Amplicon Length (bp) |
|---|---|---|---|
| Sqr-F Sqr-R | Sulfide quinone reductase | GCTCGGCAGCCTCAATAC GGTCGGACGGTGGTTACTG | 136 |
| Sor-F Sor-R | Sulfur oxygenase reductase | AAGCCCGTGCCTAAAGTG CTGCCATAGTTGGTGTTGT | 266 |
| DoxD-F DoxD-R | Thiosulfate:quinone oxidoreductase | CATCCCAGGACTCCACAA GTCGCCACCTATTCTTACTATC | 223 |
| TetH-F TetH-R | Tetrathionate hydrolase | TGAAAGACACGCTACCCG GGCCGCTCAATGATAACC | 270 |
| HdrA-F HdrA-R | Heterodisulfide reductase A | CCGATTTGAAGGTGAAGC CGGTTGCGACCATCTGTT | 185 |
| HdrB-F HdrB-R | Heterodisulfide reductase B | GTGGACCAGCGGGAAGAA TACCACGGCTCTGGCATCG | 126 |

**Table 1.** *Cont.*

| | | | |
|---|---|---|---|
| HdrC-F HdrC-R | Heterodisulfide reductase C | TATTGAGTTTGGTCGCATTG CCCTTGGACAGACGCTTT | 114 |
| soxA-I-F soxA-I-R | Sox system related protein A | GCTCAGTCAGGGTAAGGC GACAACTATTCAAACGCATC | 161 |
| soxB-I-F soxB-I-R | Sox system related protein B | GCGTATTACCGATTTGCG GGATTACCGGCCATGTTT | 198 |
| soxX-I-F soxX-I-R | Sox system related protein X | GCAGGGTAATTGTTTGGC CATATTGATGTGCGGGAT | 163 |
| soxY-I-F soxY-I-R | Sox system related protein Y | GGAATGTCAGCAGTGGGTAT TTCTCCGCTATGGTTGGT | 203 |
| soxZ-I-F soxZ-I-R | Sox system related protein Z | AAGCGGGCAAGTTGATTC CGTATTGTCTTTCCAGGTC | 173 |
| Rhd-F Rhd-R | Rhodanese | GTGGTCCTGCTTACCCTCAA GCCCGATAATATCCTGCTACTG | 130 |
| CysA-F CysA-R | sulfate/thiosulfate import ATP-binding protein | GCCCCATGCAATTCAGTAGT GCTGAAGGAGCGTTGTAAGC | 208 |
| CysB-F CysB-R | cys regulon genes regulator | AGGCTTCATGCTTGACCAGT GTTGTACGCCGACAATCTGA | 197 |
| serine O-F serine O-R | serine O-acetyltransferase | GTGTGCATGCCCTGTTTATG GTCACCAATTTCTGCCGTCT | 197 |
| HMT-F HMT-R | heavy metal transport/detoxif -ication protein | GGCACTTCGGGTCCTCTATT GACGATGTGATGTTCGGTTG | 119 |
| CzcA-F CzcA-R | Cobalt-zinc-cadmium resistance protein CzcA gene | GCAGATTCCCCTCGCACAGT CCAATACTCGTCCCCGGTTT | 123 |
| APM-F APM-R | arsenical pump membrane protein gene | GTTGGGTGCTTGTATTGCTG AAAAGTCGCTGTGGGTGAAA | 112 |
| Copper-F Copper-R | copper resistance gene | GCAAGGACTTACAGGGCACG TGACCATACGATTGATTAGACGAT | 152 |
| CDF-F CDF-R | cation diffusion facilitator family transporter gene | TCCGCTGCTCAGTGTCTCC GCACCACCCTCTTCGTCA | 129 |
| RND-F RND-R | RND family efflux transporter membrane fusion protein subunit gene | AAAGTGTCGCAACCAGTCG CAGCGGGAACCAGATAGTGT | 129 |
| CzcAEP-F CzcAEP-R | CzcA family heavy metal efflux pump | ATCGTGCGTTGGTGTATGGA CGAGATGTCGGGGAGTGCTT | 117 |
| CDB-F CDB-R | Adenosine triphosphate(ATP)-binding-cassette (ABC) transporter ATP-binding protein | TTCCTCTGGGCATCAAACAA TCATCCGTGAAATGGGTGGT | 184 |
| CDC1-F CDC1-R | ABC transporter permease, putative | TAGCCATTGCTTTTGTCCTG CTGGTTTCTGCGGTGGGTTG | 118 |
| CDC2-F CDC2-R | ABC transporter permease, putative | TGTTCTGGTTTCAGGTGCCC CCTTGCTGTTGAGTGACCGA | 127 |
| CDE-F CDE-R | MerR family transcriptional regulator | CAAGTCTGCTCAGCACCTCA CTGACTGCAGGAACGAATG | 201 |

## 3. Results

### 3.1. Growth of At. thiooxidans A01 under Different $Ni^{2+}$ Concentrations

The growth characteristics of *At. thiooxidans* A01 under the influence of $Ni^{2+}$ in different concentrations from 0 to 600 mM was studied with regard to media pH changes and densities of bacterial cell cultures (Figure 1); 600 mM $Ni^{2+}$ completely inhibited the growth and sulfur metabolism of *At. thiooxidans* A01, and longer lag phase and shorter exponential phase were observed with the addition of $Ni^{2+}$. The maximum cell density was $2.02 \times 10^8$ cells/mL in blank, while in $Ni^{2+}$ varied from 50 to 600 mM; the maximum cell densities were $1.16 \times 10^8$, $9.6 \times 10^7$, $8 \times 10^7$, $4.76 \times 10^7$, $2.38 \times 10^7$, $1.19 \times 10^7$, $1.02 \times 10^7$ cells/mL, respectively. The final pH value gradually reduced and varied from 0.32 to 1.86 with the increase of $Ni^{2+}$, and up to 600 mM $Ni^{2+}$ no notable decline could be observed. It is clear that the presence of 600 mM $Ni^{2+}$ completely inhibited the growth and sulfur metabolism of A01.

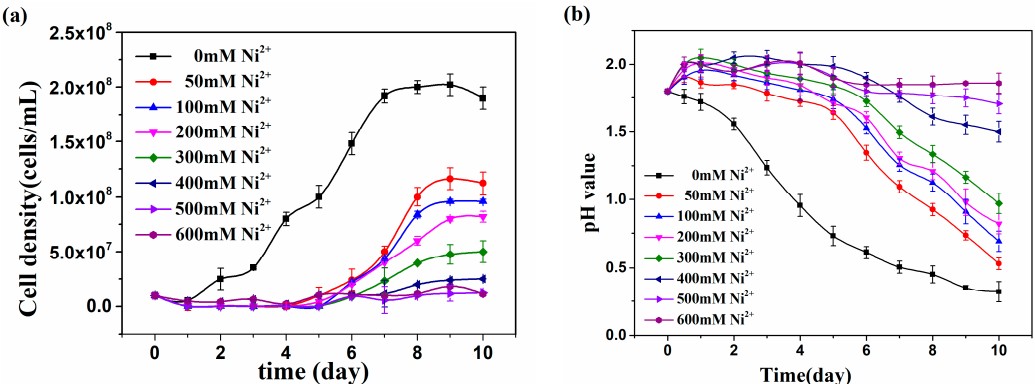

**Figure 1.** The cell density (**a**) and pH value (**b**) of *Acidithiobacillus thiooxidans* A01 under different $Ni^{2+}$ concentrations.

### 3.2. The effect of Ferric Iron on Growth of At. thiooxidans A01 under $Ni^{2+}$

$Ni^{2+}$ and $Fe^{3+}$ are often encountered in bioleaching systems, so the effect of $Fe^{3+}$ in different concentrations from 0 to 50 mM was investigated under the influence of 600 mM $Ni^{2+}$ (Figure 2). This showed that *At. thiooxidans* A01 could tolerate a high concentration of $Ni^{2+}$ with the addition of $Fe^{3+}$. The maximum cell densities increased with the concentration of $Fe^{3+}$ and were $4.91 \times 10^7$, $1.04 \times 10^8$, $1.84 \times 10^8$ cells/mL respectively, when *At. thiooxidans* A01 was exposed to 0.5, 2.5, and 5 mM $Fe^{3+}$. But further increases of $Fe^{3+}$ showed no further effect on growth. With the addition of 5 mM $Fe^{3+}$, the pH values decreased from 1.80 to 0.95 quickly after 2 days. All the Eh decreased to 450 mv in 10 days. The results indicated that 5 mM $Fe^{3+}$ restored the inhibition of $Ni^{2+}$ on *At. thiooxidans* A01. There is no doubt that $Fe^{3+}$ played a vital and positive role in the growth and metabolism of *At. thiooxidans* A01 under high concentration of $Ni^{2+}$.

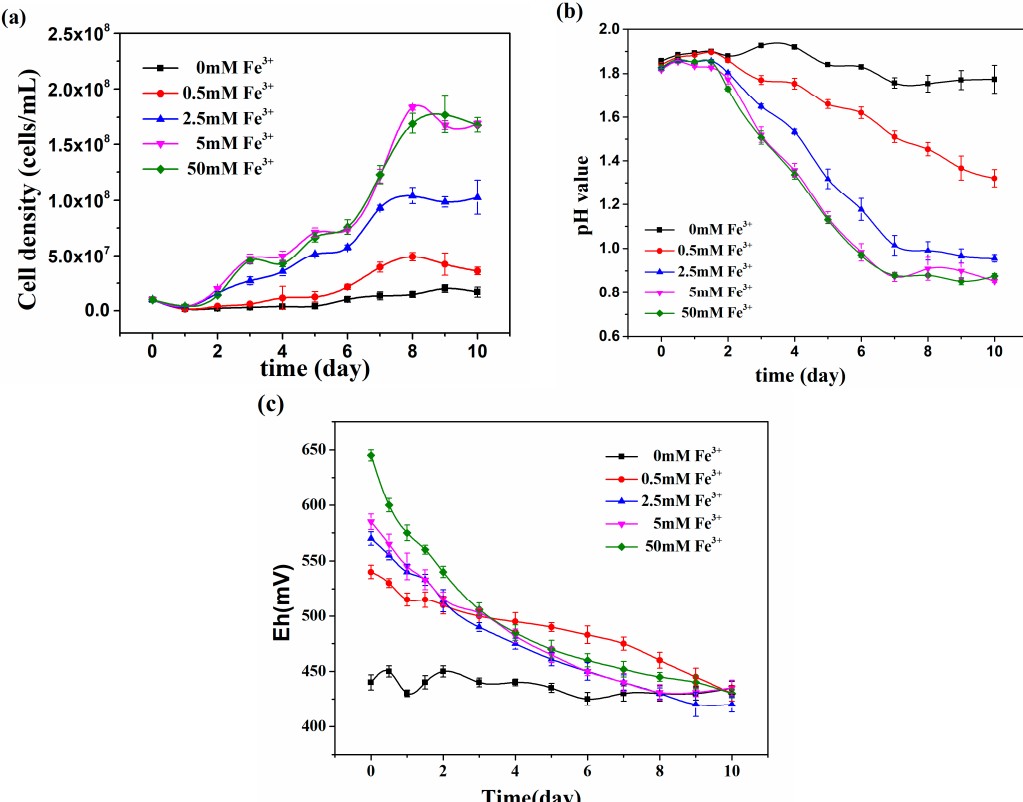

**Figure 2.** The effect of ferric iron on the cell density (**a**), pH value (**b**) and Eh (**c**) of *At. thiooxidans* A01 under the influence of 600 mM $Ni^{2+}$.

### 3.3. Growth of At. thiooxidans A01 under Shock Treatment of Individual and Joint Ni$^{2+}$ and Fe$^{3+}$

*At. thiooxidans* A01 at exponential phase (108th hour) was shocked with individual and joint 600 mM Ni$^{2+}$ and 5 mM Fe$^{3+}$ (Figure 3). The bacterium fell into recession after Ni$^{2+}$ shock and cell density decreased from $1.00 \times 10^8$ cells/mL to $6.00 \times 10^7$ cells/mL in 24 h; however, after joint Ni$^{2+}$ and Fe$^{3+}$ shock the cell density restarted increasing until the cell density reached $1.20 \times 10^8$ cells/mL after 24 h. The pH value was 0.77 at 108th hour without shock, while in the presence of Ni$^{2+}$ and in the presence of Ni$^{2+}$ and Fe$^{3+}$ the pH value increased from 0.77 to 0.98 in 2 h subsequently down to 0.76 and 0.55 at 240th hour, respectively. It can be seen that the addition of Fe$^{3+}$ would alleviate the shock of 600mM Ni$^{2+}$ on *At. thiooxidans* A01.

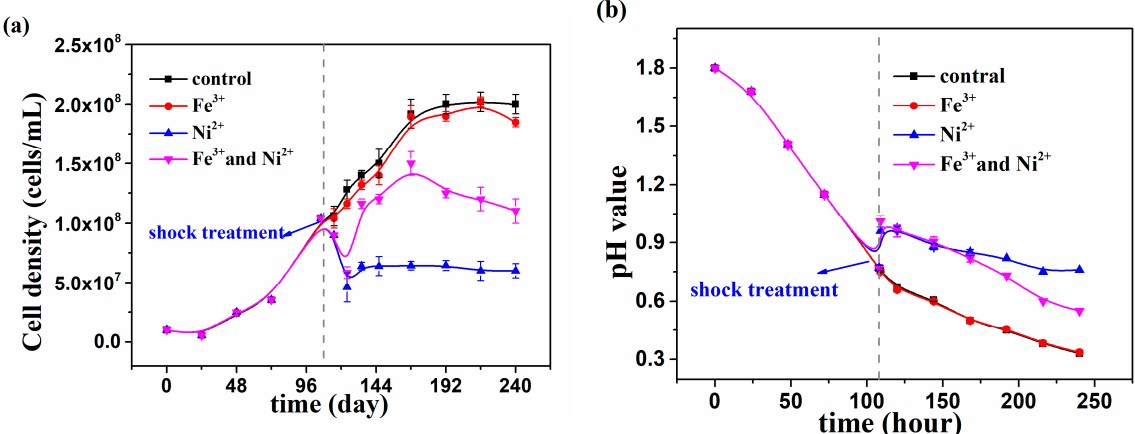

**Figure 3.** The cell density (**a**) and pH value (**b**) of *At. thiooxidans* A01 under shock treatment of individual and joint Ni$^{2+}$ and Fe$^{3+}$.

### 3.4. Functional Gene Expression of At. thiooxidans A01 by Single or Joint Ni$^{2+}$ and Fe$^{3+}$ Shock

The expression of metal resistance-related and sulfur metabolism genes was analyzed at transcriptional level by real-time PCR to investigate the effect of Fe$^{3+}$ on *At. thiooxidans* A01 after the shock of Ni$^{2+}$.

The expression of metal resistance-related genes in the 10th min, 30th min, 60th min, 120th mins after shock treatment is shown in Figure 4. The expression of metal resistance-related genes was more active in the presence of Ni$^{2+}$ and Fe$^{3+}$. At first, the expression of genes encoding adenosine triphosphate(ATP)-binding-cassette (ABC) transporter ATP-binding protein (CDB) in the presence of Ni$^{2+}$ and Fe$^{3+}$ was threefold higher than that in the presence of Ni$^{2+}$ and up regulated in the 30th min. The same regulation occurred in the expression of genes encoding CzcAEP protein. During the 30th to 60th min, the expression of most tested genes was consistently upregulated, and the expression in the presence of Ni$^{2+}$ and Fe$^{3+}$ was higher than that in the presence of Ni$^{2+}$. In particular, the expression of genes encoding RND, copper, CDB, and ABC transporter-related proteins CDC$_1$ and CDC$_2$ was more than twofold higher than that in the presence of Ni$^{2+}$. These data indicated that transporter genes were overrepresented with additional Fe$^{3+}$. In 120th min, the expression of most genes was down regulated. But in the presence of Ni$^{2+}$, the expression of genes encoding CzcAEP was four-fold higher than that in the presence of Ni$^{2+}$ and Fe$^{3+}$.

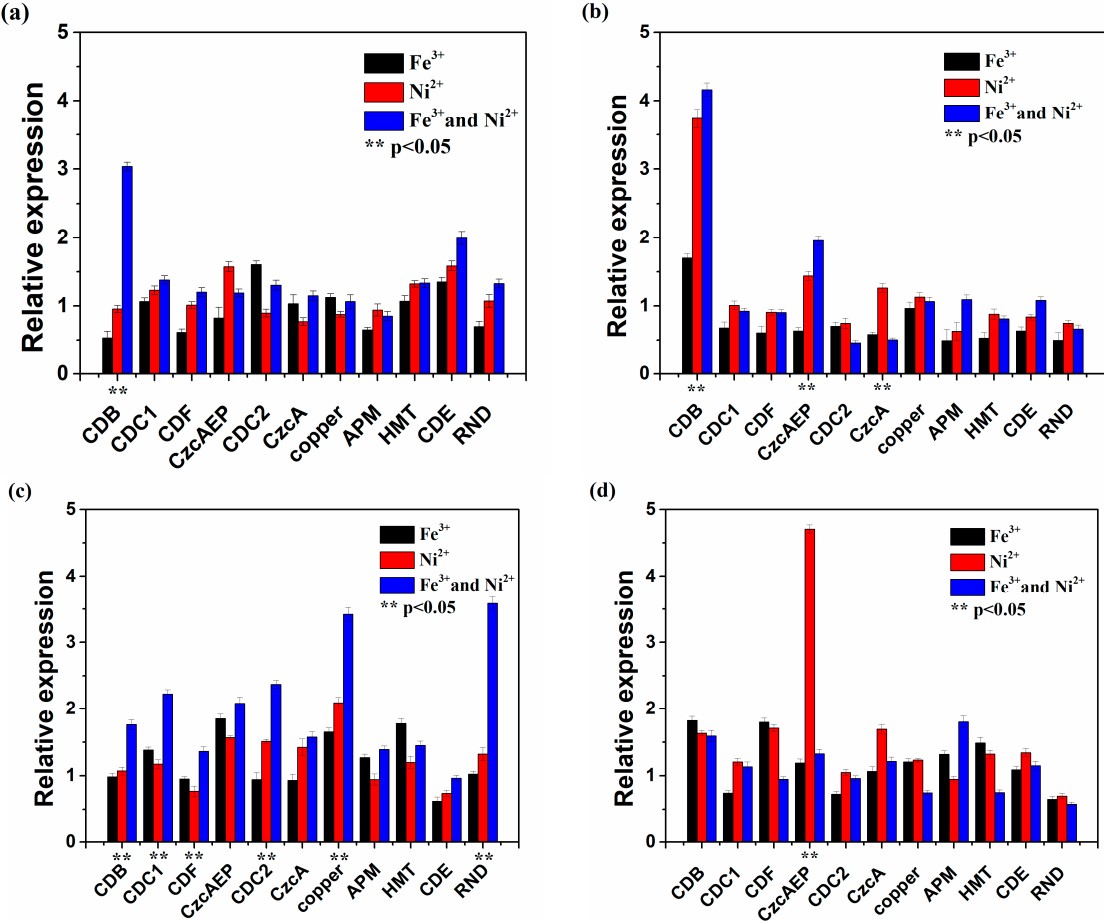

**Figure 4.** The expression of metal resistance-related genes in *At. thiooxidans* A01 in 10th min (**a**), 30th min (**b**), 60th min (**c**), 120th min (**d**) after single or joint $Ni^{2+}$ and $Fe^{3+}$ shock.

The expression of sulfur metabolism-related genes after single or joint $Ni^{2+}$ and $Fe^{3+}$ shock was shown in Figure 5. In the 10th min, most genes in the presence of $Ni^{2+}$ and $Fe^{3+}$ had higher expression, but from 30 min to 120 min after a single $Ni^{2+}$ shock, the expression of most investigated genes was higher. It is obvious that $Fe^{3+}$ would only promote the expression of sulfur metabolism-related genes at the early stage. It is also worth mentioning that the expression of *cysA* in the presence of $Ni^{2+}$ and $Fe^{3+}$ was much higher than that in the presence of $Ni^{2+}$ or $Fe^{3+}$, after 30 min the expression of most genes was upregulated in the presence of $Ni^{2+}$, especially *serinO*. Compared with the other genes, the expression of *soxZ-1*, *rhd*, *cysA*, *cysB* and *serinO* was higher in the presence of $Ni^{2+}$ and $Fe^{3+}$ and in the presence of $Ni^{2+}$ in the 60th min. In the 120th min, the expression of *rhd* in the presence of $Ni^{2+}$ was six-fold higher than that in the presence of $Ni^{2+}$ and $Fe^{3+}$. The results indicated that these sulfur metabolism-related genes were vital to maintain growth activity under high nickel shock.

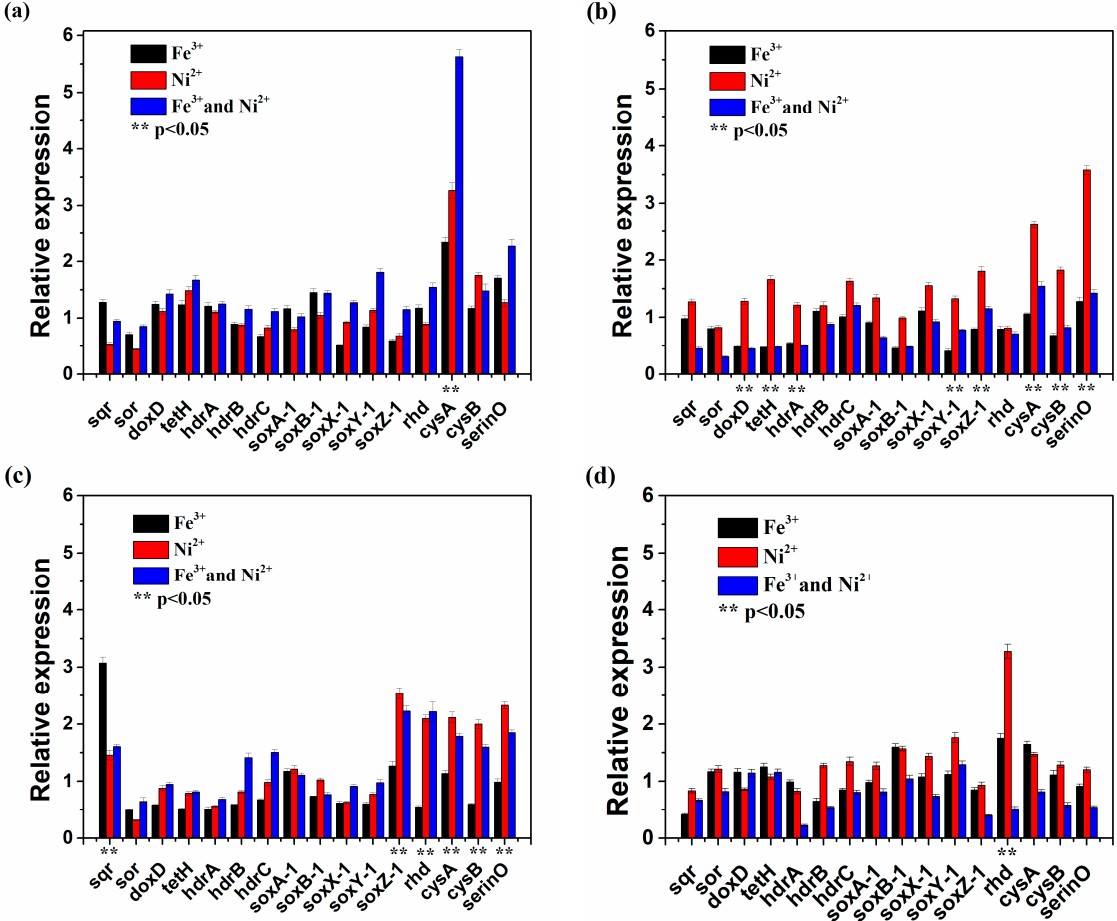

**Figure 5.** The expression of sulfur metabolism-related genes in *At. thiooxidans* A01 in 10th min (**a**), 30th min (**b**), 60th min (**c**), 120th min (**d**) after single or joint $Ni^{2+}$ and $Fe^{3+}$ shock.

## 4. Discussion

*At. thiooxidans* A01 plays a key role in bioleaching processes and commonly thrives in the Earth's most metal-rich environments [21]. Previous research indicated that metal ions can affect the heavy metal resistance of bacteria [22,23], so we investigated the effect of $Fe^{3+}$ on the growth and related genes expression of *At. thiooxidans* A01 under the stress of $Ni^{2+}$. *At. thiooxidans* A01 grows slowly and with a reduced ability to oxidize elemental sulfur in the presence of 300 mM $Ni^{2+}$ [18], but we found that trace amounts of $Fe^{3+}$ (0.5 mM) facilitated the growth and sulfur metabolism of *At. thiooxidans* A01 in the presence of 600 mM $Ni^{2+}$ and it could not further alleviate the toxicity when $Fe^{3+}$ was more than 5 mM, indicating that $Fe^{3+}$ may not weaken the toxicity through the complex with nickel.

Iron participates in a variety of essential biological functions, including metalloregulation, structural stabilization, electron transfer, substrate/cofactor coordination and catalysis [24,25]. Previous studies have shown that growth perturbations induced by metal ion stress can be attributed to disruption of cellular metal ion homeostasis, commonly resulting from protein mismetallation [26,27]. In this study, high concentrations of $Ni^{2+}$ inhibited the growth and sulfur metabolism of *At. thiooxidans* A01. $Ni^{2+}$ could improperly bind to metalloproteins and cause oxidative stress (production of reactive oxygen species, including free radicals and peroxides), which damage major molecular components. The iron-sulfur clusters were the primary intracellular targets of metal ion toxicity. Higher concentrations of metal ion led to total disintegration of the Fe-S cluster in central catabolic and biosynthetic pathways, evidently displacing iron atoms [24,28,29]. Some iron-containing enzymes are inhibited by nickel, with some shown to have nickel replacing the active metal [26]. To avoid such a process, the intracellular concentration of metal is tightly regulated and bacteria require precise

homeostatic mechanisms to balance the uptake and storage of different metals. Our results revealed that the external supplementation of ferric iron in the media could correct the effects of $Ni^{2+}$ toxicity. This can suggest that there were competitive interactions between $Fe^{3+}$ and $Ni^{2+}$, which resulting in the reconstruction of the metalloproteins and restore the growth of *At. thiooxidans* A01 under 600 mM $Ni^{2+}$. A similar experiment implied that ferrous iron gave significant protection to *At. caldus* BC13 from lead and zinc toxicity [30].

*At. thiooxidans* A01 grows and survives by autotrophically utilizing energy derived from the oxidation of elemental sulfur and reduced inorganic sulfur compounds (RISCs). In the presence of $Ni^{2+}$ and $Fe^{3+}$, the expression of most sulfur metabolism-related genes was higher compared to the control without $Ni^{2+}$. Sulfur metabolism-related genes including *sox* and *rhd* were upregulated after a shock with high nickel concentration. *Sox* complex encodes the sulfur oxidation protein, and it can directly catalyze thiosulfate to sulfate [21], and *rhd* encodes a mitochondrial enzyme that detoxifies cyanide ($CN-$) by converting it to thiocyanate ($SCN-$) and yielding to the formation of sulfite [19]. Similar to our results, $Zn^{2+}$ and $Cd^{2+}$ increased the expression of genes involved in the sulfur assimilation pathway in *At. ferrooxidans* [31]. In addition, $Cd^{2+}$, $Zn^{2+}$, and $Cu^{2+}$ exposure increased the expression of a putative high-affinity sulfate transporter gene and root sulfate uptake capacity in maize roots [32]. In the presence of $Ni^{2+}$ and $Fe^{3+}$, the expression of most sulfur metabolism-related genes was higher and then downregulated compared to the individual nickel shock. This indicated that *At. thiooxidans* A01 could respond rapidly to the disturbance. Some bioleaching organisms utilize only $Fe^{3+}$ as an electron acceptor although most of the sulfite acceptor oxidoreductases were shown to use c-type cytochromes as electron acceptors [33]. Additional $Fe^{3+}$ is probably involved in sulfur assimilation and uptake in *At. thiooxidans* A01.

It is worth noting that *cysA*, *cysB* and *serineO* were upregulated in the presence of $Ni^{2+}$ and $Fe^{3+}$. Sulfate/thiosulfate import ATP-binding protein (cysA) and transport sulfate as a sulfur source for cysteine biosynthesis [34]. Moreover, serine O-acetyltransferase (serine O) catalyze the reaction: L-serine + acetyl-coenzyme A $\leftrightarrows$ O-acetyl-L-serine + coenzyme A [35], subsequently, the O-acetyl-L-serine can be converted into organic sulfur molecules cysteine [36]. The cysteine is the precursor of well-known chelators of heavy metals such as metallothioneins, glutathione and phytochelatins [37]. Some studies indicate that appropriate amounts of L-cysteine can significantly improve the bioleaching of Ni–Cu sulfide and pyrite [38,39]. Chelation of heavy metals is a ubiquitous detoxification strategy in a wide variety of organisms. Our results indicated that $Fe^{3+}$ promotes the expression of sulfur assimilation-related genes that have an impact on the survival of *At. thiooxidans* in high concentrations of nickel by producing precursor of chelators of heavy metals.

Microorganisms have evolved ways to protect themselves from metal overload. A common microbial response to elevated concentrations of toxic metal is to synthesize efflux system, thus reducing the internal concentration of metal [7]. In the presence of $Ni^{2+}$ and $Fe^{3+}$, genes encoding a series of RND proteins including CzcAEP and the copper resistance gene *afe_1073* showed high expressions. The RND protein family is a huge superfamily involved in exporting superfluous heavy metal including $Ni^{2+}$ and referred to as CBA efflux systems. CzcAEP is part of the CzcCBA complex (RND protein) that mediates heavy metal resistance [40–42]. Iron is an essential metal for cellular functions under high nickel stress. It was speculated that $Fe^{3+}$ promoted the expression of RND-related proteins driven by the cation-proton antiporter to facilitate nickel transport from periplasm to the outside of membrane. Similarly, extracellular zinc acted in a dose dependent manner to competitively inhibit manganese uptake by *Streptococcus pneumonia* [43]. Zinc stress induces copper depletion in *Acinetobacter baumannii* [44].

The ABC transport system is another basic defense against heavy metals [45]. Genes encoding ABC transporter-related proteins (CDB, $CDC_1$, $CDC_2$) were upregulated with additional $Fe^{3+}$. They were shown to increase resistance to nickel in *L. ferriphilum* YSK and *Escherichia coli* strains [41,46]. In addition, extra metal ions including $Ni^{2+}$, $Cu^{2+}$ and $Mn^{2+}$ induced siderophores' production under an iron-limited condition [47]. Competition experiments between iron and other metals for

pyoverdine showed a clear preference for iron [48]. Once iron is chelated by siderophores, it was delivered via various types of transporters including the ABC transport system to sequester this metal [2]. Thus, iron can possibly keep $Ni^{2+}$ out of cytoplasm by completion. These data indicated that *At. thiooxidans* A01 might acquire specialized mechanisms to sense, transport and maintain $Fe^{3+}$ within physiological concentrations and to detoxify non-essential metal $Ni^{2+}$.

## 5. Conclusions

Collectively, this work reveals the resistance strategies utilized by *At. thiooxidans* A01 to survive $Ni^{2+}$ stress with additional $Fe^{3+}$, allowing it to thrive in diverse environments. Our results indicated that $Fe^{3+}$ restored the growth of *At. thiooxidans* A01 in the presence of $Ni^{2+}$ by alleviating $Ni^{2+}$ toxicity, exporting intracellular nickel ions, and producing chelators of heavy metals. However, further metallomics, biochemistry and biophysics analysis such as cellular elemental distribution, enzymatic activity, spectroscopy, thermodynamics, and redox chemistry will be performed to decipher the precise mechanism.

**Author Contributions:** Conceptualization, Y.L., A.C., Y.X. and X.H.; Methodology, A.C. and Y.X.; Resources, G.Q., X.L., Y.L., H.Y. and H.L.; Data Curation, A.C., Y.L., K.Z. and Y.X.; Writing—Original Draft Preparation, A.C.; Writing—Review and Editing, A.C., Y.X., X.H. and Y.L.; Project Administration, Y.L.

**Funding:** This research was funded by [National High Technology Research and Development Program of China] grant number [2012AA061502], [National Key Research and Development Program of China] grant number [2016Y FB0101310] and [the National Natural Science Foundation of China] grant number [31570113].

**Acknowledgments:** We are grateful for constructive comments by three reviewers.

**Conflicts of Interest:** The authors declare no conflict of interest.

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
