# Peer review of "Responses of Acidithiobacillus thiooxidans A01 to Individual and Joint Nickel (Ni2+) and Ferric (Fe3+)"

_minerals, doi:10.3390/min9020082_

Round 1
Reviewer 1 Report
The authors describe here a growth kinetic of the extreme acidophile Acidithiobacillus ferrooxidans under the influence of metal shock. While the overall experimental and scientific level of their work is on a suitable level, the style and manner of the representation do not permit a sufficient understanding of their work. A check of a native speaker is strongly recommended before resubmission of the revised manuscript. The topic of their investigations and the obtained results are quite interesting for the scientists in this filed, however, the overall style of the representation needs to be dramatically improved. Additionally, this manuscript is lacking an interesting conclusions, which can be developed based on their results. The line with synergetic interactions between Fe3+, Ni2+ and their microorganism needs to be very well developed. I understand, that the current methodological level applied by the authors may not provide data to understand the complexity of Ni-Fe microbial chemistry. At least, the authors may suggest an appropriate technology how to decipher Ni-Fe synergy in their case. For instance, analytical biochemistry and biophysics techniques (spectroscopy, cellular elemental distribution, redox chemistry) would provide more information to understand the positive effect of combined Fe, Ni-stress.
Quite a long list of detailed “minor” suggestions:
Title: Please give a full name of your microorganism in the title: Acidithiobacillus ferrooxidans
L8: Here At. thiooxidans should be Acidithiobacillus ferrooxidans, as this is the first time mentioned. Further, At. thiooxidans can be used everywhere, but no abbreviators like A01 may be applied in the scientific type of a publication!
L8-9: please make everything in the same font style and size.
L9: should be “sensitivity to nickel”.
L9-11: Here is your sentence “The effect of ferric iron on the growth of At. thiooxidans under the stress with 600 mM/L nickel has been studied, and the up-regulation of the genes related to sulfur oxidation and metal transport was observed with RT-PCR.”
L11-14: please rephrase this; it is too long for 1 sentence and hard to see any sense the way the phrases are connected.
L16: should be bacterium.
L21: Any should be with a capital letter?
L26: the line begins with a point??
L29: should be “dioxygenases”, there is something wrong with “sulfite oxidize”
L30: should be “and indirectly to its …”
L31: nickel resistant bacterium
L44: please rephrase, it makes no sense “that Ni(II) and Co(II) in combination make the toxicities were significantly reduced”
L45: should be At. caldus
L50: should be “At. thiooxidans is a conspicuous dweller”.
L53: should be At. thiooxidans
L57: should be At. thiooxidans and stress with what??
L58: should be At. thiooxidans A01
L58-59: it makes no sense, please rephrase “under the stress of Ni 2+ to clarified the mechanism at transcription level”.
L64: please rephrase “which containing”
L69: should be “incubation”
L74: was performed in triplicate.
L79: was performed or conducted to explore…
L70, L84: how S0, ferric sulfate and nickel sulfate were sterilized? It is stated in L 70 that you applied UV-treatment. I doubt that UV sterilization provides you the conditions to maintain a pure culture of your microorganisms. There are plenty of extremophiles including acidophiles that can resist UV radiation. It can be very well that you were cultivating not only At. thiooxidans.
L79-85: which were the initial and final cell densities of the experiments? At “0” time point, shock with metals and harvest time points?
L95: the genome has been sequenced..
L97: genes were predicted based on…
L98: are shown
L103: please rephrase, should be “under the influence of Ni2+ in different concentrations from 0 to 600 mM/L”, not under the pressure…
L106: please begin a sentence with a capital letter.
L109: should be “reduced and varied”
L111: At. thiooxidans should be in italic, “concentrations”
L113: “the effect of Fe3+ in different concentrations from 0 to 50 mM/L was investigated…”
L114: should be “under the influence of 600 mM/L Ni2+”;
L114, 116, 121, 122, 134, 139, 169, 170, 177, 179, 190: please use the full name At. thiooxidans A01, it is not possible to use only A01
L117-120: please split and rephrase this sentence, you can easily generate 3 sentences out of it to make it understandable. At. thiooxidans should be in italic.
L122-123: one cannot say this from your result, but you can clearly state that “synergic interaction between Ni2+ and Fe3+ played a vital positive role in At. thiooxidans A01 adapting…”
L128: should be “the bacterium”; please rephrase “advanced into recession”
L135, 151, 163: At. thiooxidans should be in italic
L141: are shown; what does it mean “Fig. 4, 30 min.”??
“mental resistance-related genes” do you mean metal-resistance???
L153-154: something one: either shock or treatment, but not both together…
L155: need a space after 30; should be “most investigated”??? “It is obviously”, please use present tense to make your statements and split it to a separate sentence.
L157: It is worth…
L138-150 and L153-162: these paragraphs require an intensive correction of a native speaker. There is just a number of phrases not connected with each other. This text requires a sufficient rephrasing. I can only recommend do not to use “under N”, “under NF” and so on, but properly, in a scientifically correct way to decipher in the text under which condition what is observed. One can simply say “in the presence of Ni”, “in the presence of Ni and Fe”, etc.
L169: no need of space “Ni 2+ ”.
L170: should be “grows slowly and with a reduced ability to oxidize elemental sulfur in the presence of 300 mmol/L Ni2+ ”
L172: what is this “A.t A01”??? should be “in the presence of 600 mM/L Ni2+”
L174: should be A. caldus
L175: should be plural, microorganisms
L176: which heavy mental??
L177: what is this “A01in”??? Probably should be “genes…were” here…
L180: “why the present of Fe3+ made a shorter lag phase” – this is not possible to understand!
L180: CDB was shown to transport…
L181: Please separate sentences here, rephrase, it is not possible to understand “in Escherichia coli strains AtmA (the ABC-type transport system) increased resistance that carry deletions of the genes for other nickel transport systems”.
L185: acquired instead of “got”
L186: which exactly gene of Czc complex or the whole complex itself?
L189: what is this “A.t A01”??? you cannot use this type of abbreviators in a scientific manuscript.
L189-190: please formulate a scientifically and grammatically correct sentence here.
L201-202: should be “after a shock with high Ni concentration”
L202: should be “encodes”, no space after protein. Not “could”, it can directly catalyze…
L204: “and yielding to the formation of sulfite”.
L205: should be At. thiooxidans.
L206: should be “increased the expression of ZmST1;1, which is a putative… ”
Line208 and everywhere through the text: should be “under the conditions of Ni and Fe combined stress, the expression… ”
L209: it is worth to note
L213: “A[33],subsequently,” needs to be separates by coma
L213: “cysteine[34].The” – the same
L216: “sulfid e” ???
L218: perhaps “in the production of a number of compounds… ”???
L219: “… bonds upon bacterial attachment to the mineral surface…”
L217: in the presence of A. caldus
L221: “A.t A01”???
L219-221: here is your corrected sentence: “Our results indicate that Fe3+ promotes the expression of sulfur assimilation related genes including cysA, cysB and serine O that have an impact on the survival of At. thiooxidans in high concentrations of nickel.”
L221-223: this reference is not really relevant to your investigations, you do not work on single protein level, therefore it is not possible to compare, please remove this sentence.
L223-224: Here is your sentence “Our investigations suggests that microbial mediated sulfur oxidation can be influenced by multitudinous external factors of a chemical nature”.
L224-226: This cannot be concluded based on your results; one would need to show a functional respiration and a growth of your microorganism in the presence of Fe3+ and in the absence of any sulfur source in order to talk about such possibilities.
Author Response
Dear Reviewers:
Thank you for the reviewers comments concerning our manuscript entitled “Responses of Acidithiobacillus. thiooxidans A01 to Individual and Joint Nickel Ni(II) and Ferric Fe (III) ”. These comments are all valuable and very helpful for revising and improving our paper, as well as the important guiding significance to our researches. We have studied comments carefully and have made correction which we hope meet with approval.
Once again, thank you very much for your suggestions.
Reviewer 2 Report
The manuscript investigates the expression of genes related to metal resistance and sulfur metabolism by the strain At. thiooxidans A01 at different concentrations of Fe and Ni ions. The authors propose that the presence of Fe influence the Ni resistance of the cells caused by an upregulation of genes. The authors used a model system and studied the influence of Fe, Ni and Fe/Ni on gene expression using RNA analyses. Principally the article is suited for the journal but there are some major issues that should be considered:
Abstract: gene expression was studied; the proteins were not studied; conclusions should be made more carefully
Chapter 3.3: please explain the abbreviations F, N, NF
Chapter 3.4: the transporters are a resistance mechanism; principally it could be expected that they are upregulated in the presence of toxic Ni; why are they upregulated in the presence of Ni/Fe in comparison to Ni? Please include it in the discussion
Discussion: mechanisms/meaning of upregulated genes should be discussed more intensively; the structure of the discussion should be improved, currently it is confusing; e.g. discussion of genes related to sulfur metabolism should concentrate on metal resistance; further I recommend to concentrate the discussion on few genes showing a significant difference in gene expression (e.g. CDB, copper, RND CzcAEP, cysA); in most cases the difference is not significant or the gene expression in presence of Fe or Ni and Fe is similar.
Gene expression is only one indicator for inducing stress tolerance; the presence of metals influences also the activity of enzymes, e.g. transporters; increased gene expression could be also a response on the inactivation of the enzymes; the enzyme activities were not investigated. These issues should be discussed. A comparison of protein profiles would be also nice.
There are many typos and grammatical errors throughout the text. I should be revised by a native speaker.
Author Response
Dear Reviewers:
Thank you for the reviewer comments concerning our manuscript entitled “Responses of Acidithiobacillus. thiooxidans A01 to Individual and Joint Nickel Ni(II) and Ferric Fe (III) ”. These comments are all valuable and very helpful for revising and improving our paper, as well as the important guiding significance to our researches. We have studied comments carefully and have made correction which we hope meet with approval.

Round 2
Reviewer 1 Report
Thank you very much for the corrected and improved manuscript. However, the authors have not carefully corrected all the addressed comments; the manuscript still has several issues to take care of:
L20-21: either you use up regulated or over expressed, but please not “up-expressed”
L22: can be attributed, not “was attributed”.
L24: It can suggest instead of “It suggested”
L25: should be in plural “heavy metals”
L31-33: Just look carefully at these lines: after your reference “microorganisms[1].” you have a full stop of your sentence “.” Next sentence begins “Any of heavy metals..”. “Any” should be with a capital letter. Hope it helps.
L36: “[2] .” No space in between please
L39: There is something strange after “dioxygenase、”, it should be just a coma.
L50: Cupriavidus metallidurans in italic please
L51: should be plural “proteins”?
L61-61: Please use uniformly through your whole manuscript one way to present the oxidation states of your metals: Ni2+, Fe3+, Co2+, given as a superscript, instead of Ni (II) and Co (II) please use Ni2+, Co 2+, and everywhere else.
L64: should be against lead, not “from lead”
L66: heavy metals please!!!!!!!!!!!!!!!!!!!
L82-83: “medium, which contained”
L89, L104: you need to say that you applied UV sterilization for your metals. And please provide in the manuscript all the details of this sterilization: how long you applied UV, which kind of UV exactly, settings (power, spectral range, etc.).
L102: What is it “the 108th ”, 108 hours?
L105: Please provide the cell densities at the harvest time points.
L124: plural please “in different concentrations”.
L125: instead of “bacterial concentrations” please use “densities of bacterial cell cultures”
L137: “was exposed”
L138: “no further effect” instead of “no more effect”.
L149: What is it “the 108th ”, 108 hours?
L164: “is shown” instead of “are shown”
L166: “was threefold higher” instead of ”were threefold higher”
L172: should be “transporter genes” instead of “transporter proteins”; “were overrepresented” instead of “were more thriving”.
L172-173: should be “the expression of most genes was down-regulated”
L184: should be “It is…”
L187: should be “was up regulated”
L210: metals in plural please
L211: what do you mean “Iron metals”? should it be perhaps simply “Iron participates …?
L216: should be “Excess of ferric iron”
L219-220: “Further biochemistry and biophysics analysis will perform to decipher the mechanism”. I am afraid it is not sufficient to provide this kind of general sentence. Please refer to the previous comments and suggest here which exactly appropriate techniques you could recommend to use in order to decipher Ni-Fe synergy.
L233: should be “to facilitate”
L253: should be “ were up regulated under the combined stress with Ni and Fe”
L261: should be “very important role in the production of a number of compounds”
L264: Please decipher who are “They”, it is not clear here, needs to be rephrased.
Author Response
Dear Reviewer:
Thank you for your letter and for the reviewers’ comments concerning our manuscript entitled “Responses of Acidithiobacillus thiooxidans A01 to Individual and Joint Nickel (Ni2+) and Ferric (Fe3+)” (ID: 398647). Those comments are all valuable and very helpful for revising and improving our paper, as well as the important guiding significance to our researches. We appreciate for Reviewers’ warm work earnestly, and hope that the correction will meet with approval.

Reviewer 2 Report
Ni(II) and ferric Fe”
The manuscript investigates the expression of genes related to metal resistance and sulfur metabolism by the strain At. thiooxidans A01 at different concentrations of Fe and Ni ions. The authors propose that the presence of Fe influence the Ni resistance of the cells caused by an upregulation of genes. The authors used a model system and studied the influence of Fe, Ni and Fe/Ni on gene expression using RNA analyses.
The authors did not upload the response to the reviewer comments, therefore it is difficult to estimate whether they addressed the comments. The manuscript, in specific the discussion, has been improved. However, there are still some issues that should be considered:
The discussion has been improved; however, the authors investigated only gene expression as one indicator for inducing stress tolerance; the authors did not investigated enzyme activities etc.; therefore conclusions should be drawn from the results more carefully (e.g. p. 10) and other mechanisms of regulating stress tolerance should be mentioned
Discussion part: The obtained results should be better related to the known literature. Currently only few lines after a long paragraph refer to own results.
There are still many typos and grammatical errors throughout the text. Some examples: l. 31: Any; l.50: strain in italic letters; l. 52: superscript charges; ll 109/110: sentence structure; l. 124: concentrations range…; ll. 181/182: are shown; l. 219: will be performed; l. 259: studies; abbreviation NF: what is NF?
Please upload the response
Author Response
Dear Editors:
Thank you for your letter and for the reviewers’ comments concerning our manuscript entitled “Responses of Acidithiobacillus thiooxidans A01 to Individual and Joint Nickel (Ni2+) and Ferric (Fe3+)” (ID: 398647). Those comments are all valuable and very helpful for revising and improving our paper, as well as the important guiding significance to our researches.
We are thinking about other mechanisms of regulating stress tolerance, such as the addition of a small amount of iron reduces oxidative stress and affects protein activity, DNA replication and transcription. You have deep and systematic studies in this area, can you provide some more detailed ideas or references?
And we are very sorry for our negligence. And we attach the last the response to the reviewer comments behind. We appreciate for Reviewers’ warm work earnestly, and hope that the correction will meet with approval.
Thank you and best regards.

Round 3
Reviewer 1 Report
Please enter your comments and suggestions for authors
Reviewer 2 Report
The authors have significantly improved the manuscript and adressed all comments.